# Consensus Panel Recommendations for the Pharmacological Management of Breastfeeding Women with Postpartum Depression

**DOI:** 10.3390/ijerph21050551

**Published:** 2024-04-26

**Authors:** Georgios Eleftheriou, Riccardo Zandonella Callegher, Raffaella Butera, Marco De Santis, Anna Franca Cavaliere, Sarah Vecchio, Cecilia Lanzi, Riccardo Davanzo, Giovanna Mangili, Emi Bondi, Lorenzo Somaini, Mariapina Gallo, Matteo Balestrieri, Guido Mannaioni, Guglielmo Salvatori, Umberto Albert

**Affiliations:** 1Italian Society of Toxicology (SITOX), Via Giovanni Pascoli 3, 20129 Milan, Italy; rbutera@asst-pg23.it (R.B.); sarah.vecchio@asl.novara.it (S.V.); lanzic@aou-careggi.toscana.it (C.L.); guido.mannaioni@unifi.it (G.M.); 2Poison Control Center, Hospital Papa Giovanni XXIII, 24127 Bergamo, Italy; mpgallo@asst-pg23.it; 3Italian Society of Psychiatry (SIP), Piazza Santa Maria della Pietà 5, 00135 Rome, Italy; zandonellacallegher.riccardo@spes.uniud.it (R.Z.C.); ebondi@asst-pg23.it (E.B.); ualbert@units.it (U.A.); 4Psychiatry Unit, Department of Medicine (DAME), University of Udine, 33100 Udine, Italy; matteo.balestrieri@uniud.it; 5UCO Clinica Psichiatrica, Azienda Sanitaria Universitaria Giuliano-Isontina, 34148 Trieste, Italy; 6Italian Society of Obstetrics and Gynecology (SIGO), Via di Porta Pinciana 6, 00187 Rome, Italy; marco.desantis@unicatt.it (M.D.S.); afcavaliere@hotmail.com (A.F.C.); 7Department of Obstetrics and Gynecology, Fondazione Policlinico Universitario A. Gemelli, 00168 Rome, Italy; 8Department of Gynecology and Obstetrics, Fatebenefratelli Gemelli, Isola Tiberina, 00186 Rome, Italy; 9Addiction Centre, Ser.D, Local Health Unit, 28100 Novara, Italy; 10Division of Clinic Toxicology, Azienda Ospedaliera Universitaria Careggi, 50134 Florence, Italy; 11Italian Society of Neonatology (SIN), Corso Venezia 8, 20121 Milan, Italy; riccardo.davanzo@gmail.com (R.D.); gmangili@asst-pg23.it (G.M.); 12Maternal and Child Health Institute IRCCS “Burlo Garofolo”, 34137 Trieste, Italy; 13Task Force on Breastfeeding, Ministry of Health, 00144 Rome, Italy; 14Department of Neonatology, Hospital Papa Giovanni XXIII, 24127 Bergamo, Italy; 15Department of Psychiatry, ASST Papa Giovanni XXIII, 24100 Bergamo, Italy; 16Ser.D Biella, Drug Addiction Service, 13875 Biella, Italy; lorenzo.somaini@aslbi.piemonte.it; 17Italian Society of Addiction Diseases (S.I.Pa.D), Via Tagliamento 31, 00198 Rome, Italy; 18Italian Society for Drug Addiction (SITD), Via Roma 22, 12100 Cuneo, Italy; 19Italian Society of Neuropsychopharmacology (SINPF), Via Cernaia 35, 00158 Rome, Italy; 20Italian Society of Pharmacology, Via Giovanni Pascoli, 3, 20129 Milan, Italy; 21Italian Society of Pediatrics, Via Gioberti 60, 00185 Rome, Italy; salvatori.guglielmo@tiscali.it; 22Department of Medical and Surgical Neonatology Ospedale Pediatrico Bambino Gesù, 00165 Rome, Italy; 23Department of Medicine, Surgery and Health Sciences, University of Trieste, 34128 Trieste, Italy

**Keywords:** breastfeeding, postpartum depression, antidepressants, anxiolytic drugs, consensus

## Abstract

Introduction: Our consensus statement aims to clarify the use of antidepressants and anxiolytics during breastfeeding amidst clinical uncertainty. Despite recent studies, potential harm to breastfed newborns from these medications remains a concern, leading to abrupt discontinuation of necessary treatments or exclusive formula feeding, depriving newborns of benefits from mother’s milk. Methods: A panel of 16 experts, representing eight scientific societies with a keen interest in postpartum depression, was convened. Utilizing the Nominal Group Technique and following a comprehensive literature review, a consensus statement on the pharmacological treatment of breastfeeding women with depressive disorders was achieved. Results: Four key research areas were delineated: (1) The imperative to address depressive and anxiety disorders during lactation, pinpointing the risks linked to untreated maternal depression during this period. (2) The evaluation of the cumulative risk of unfavorable infant outcomes associated with exposure to antidepressants or anxiolytics. (3) The long-term impact on infants’ cognitive development or behavior due to exposure to these medications during breastfeeding. (4) The assessment of pharmacological interventions for opioid abuse in lactating women diagnosed with depressive disorders. Conclusions: The ensuing recommendations were as follows: Recommendation 1: Depressive and anxiety disorders, as well as their pharmacological treatment, are not contraindications for breastfeeding. Recommendation 2: The Panel advocates for the continuation of medication that has demonstrated efficacy during pregnancy. If initiating an antidepressant during breastfeeding is necessary, drugs with a superior safety profile and substantial epidemiological data, such as SSRIs, should be favored and prescribed at the lowest effective dose. Recommendation 3: For the short-term alleviation of anxiety symptoms and sleep disturbances, the Panel determined that benzodiazepines can be administered during breastfeeding. Recommendation 4: The Panel advises against discontinuing opioid abuse treatment during breastfeeding. Recommendation 5: The Panel endorses collaboration among specialists (e.g., psychiatrists, pediatricians, toxicologists), promoting multidisciplinary care whenever feasible. Coordination with the general practitioner is also recommended.

## 1. Introduction

Any possible consensus on the topic of antidepressant and anxiolytic drug use during breastfeeding and the possible adverse effects on breastfed infants is going to be developed in an area of clinical uncertainty. Some case-reports or case-series studies have been published in recent years, but large studies are lacking. Currently, neither the European Medicines Agency (EMA) nor the U. S. Food and Drug Administration (FDA) has approved any psychotropic drug for use during breastfeeding. Consequently, the scientific interest has been focused on the pharmacokinetics of medicines during lactation.

It has been estimated that the prevalence of postpartum depression varies according to country from 5.0% to 26.32% [1] while the prescription rates of antidepressants during the postpartum period were approximatively from 2.4% to 4.1% [2]. During breastfeeding, it is crucial to address emotional disturbances as significant issues often experienced by women. Postpartum emotional disturbances are reported to be associated with discontinuing breastfeeding before 4 months [3]. While maternity blues exert a minimal impact on the psycho-affective bond between the mother and child and typically respond favorably to psychotherapy and social support, they could potentially escalate to postpartum depression necessitating treatment with antidepressants [4].

Antidepressant and anxiolytic drugs during breastfeeding may, on the one hand, constitute a risk for the newborn but they also represent a clear benefit for patients with pregnancy-onset and/or postpartum depression, and for patients who require maintenance treatment in order to sustain recovery. Clinicians should also consider that abrupt discontinuation of antidepressants may be also associated with relapses/recurrences, with potential negative outcome on newborn care.

Health professionals should then consider the risks for the mother as well as for the newborn, particularly related to: (1) untreated psychiatric disorders during breastfeeding, (2) the toxicological risk for the nursed infant, (3) the loss of breastfeeding benefits to the mother–infant dyad.

The possible harm to the breastfed infant raises concern about prescribing antidepressants such as selective serotonin reuptake inhibitors (SSRIs), serotonin and norepinephrine reuptake inhibitors (SNRIs), tricyclic (TCAs) and atypical antidepressants, and sedative medicines such as benzodiazepines (BDZs) and non-benzodiazepine hypnotics drugs (hypnotic benzodiazepine receptor agonists–HBRAs). This concern has led to immediate discontinuation of antidepressants for patients already on treatment or to not prescribing medications to patients in need, posing them at risk for suicide or attempts, which are recognized to account for approximately 20% of postpartum deaths [5]. Hence, a more detailed risk profile of all psychotropic drugs, prescribed off-label during this period, is important. The aim of our research is to provide comprehensive knowledge to guide selection of appropriate medication for women affected by depressive disorders during breastfeeding.

## 2. Materials and Methods

### 2.1. Establishing Consensus

This review allowed the development of consensus recommendations. This process has involved the scientific societies more interested in the clinical domain of postpartum depression: the Italian Society of Toxicology (SITOX), the Italian Society of Psychiatry (SIP), the Italian Society of Pharmacology (SIF), the Italian Society of Neonatology (SIN), the Italian Society of Obstetrics and Gynecology (SIGO), the Italian Society of Pediatrics (SIP), the Italian Society of Addiction Pathology (SIPaD), and the Italian Society of Drug Addiction (SITD).

The chairperson and the coordinator defined the objectives and specific areas of investigation and selected from the aforementioned scientific societies a multidisciplinary group of 16 leading Italian experts in women’s reproductive and mental health, and in infant health. None of the experts had any potential conflicts of interest or funding sources in order to ensure the experts’ transparency and credibility. The composition of the 16-member study group was as follows: 7 toxicologists and pharmacologists, 4 psychiatrists, 2 gynecologists, 1 pediatrician, and 2 neonatologists. The role of the toxicologists, the gynecologists, the pediatrician, and the neonatologist was the evaluation of the risk of adverse events in breastfed infants after the maternal use of the medicines during lactation while the psychiatrists evaluated the risks due to the psychiatric disease of the patients involved in all published articles. The team first identified the needs underlying the consensus using the methodology of the “Nominal Group Technique” (NGT).

The NGT approach Is a face-to-face group meeting process. It is a commonly used consensus method in medical research, which uses a panel of specialists to discuss and provide prompt results for researchers.

The first face-to-face NGT was held in Bergamo, 8–10 September 2022, during the 33rd Conference of the European Network of the Teratology Information Services, followed by a second step during December 2022 when the first draft of consensus was adjusted through the multidisciplinary group feedback via e-mail. During the third step, from January 2022 to December 2023, several NGT meetings were conducted in order to reach a consensus statement on the pharmacological management of breastfeeding women with depressive disorders. The following four areas of investigation were identified:Assessment of the risks associated with untreated maternal depression (depressive symptoms or Major Depressive Disorder, excluded Bipolar Disorder diagnosis) during breastfeeding;Assessment of the overall risk of adverse events in breastfed infants associated with antidepressant and anxiolytic drug use during breastfeeding;Long-term developmental outcomes of infants’ cognitive development or behavior after maternal use of the drugs during lactation;Evaluation of pharmacological treatment of opioid abuse in breastfeeding women with depressive disorders.

### 2.2. Search Strategy

An extensive review of the literature was performed to assess the risks associated with untreated maternal depression during breastfeeding and the risk of adverse effects in breastfed infants after the use of antidepressant and anxiolytic drugs during lactation. All experts who conducted the literature review had long-term experience working in the Teratology Information Services and the departments of Neonatology and Psychiatry.

The research was focused in the PubMed, EMBASE, and Cochrane library databases. Periods between 1 January 1985 and 1 January 2024 were addressed. This information enabled the group to develop a series of recommendations. To be included, studies had to fulfill the following criteria: (1) they had to include breastfeeding patients with depression; (2) they had to provide data on possible adverse effects in breastfed infants associated with drug exposure during lactation; (3) they had to provide data about the concentrations of the drugs in breastmilk and/or in the infants’ serum; and (4) they had to be a meta-analysis, systematic review of the literature, or observational study; case-reports or case-series were also included. Publications were excluded if performed in neonates in the first days of life as their drug-related symptoms are usually due to drug placental transfer or to withdrawal syndrome.

The search for relevant studies using generic keywords and MeSH terms was the following: “depression during lactation” AND “drug adverse effect” OR “neonatal/infant outcome” AND “selective serotonin reuptake inhibitors” OR “serotonin and norepinephrine reuptake inhibitors” OR “tricyclic antidepressants” OR “atypical antidepressants” OR “benzodiazepines” OR “hypnotic benzodiazepine receptor agonists” OR “opioids” AND “during breastfeeding”. Namely, the following compounds were considered: citalopram, escitalopram, fluoxetine, fluvoxamine, paroxetine, and sertraline for the SSRIs; venlafaxine and duloxetine for the SNRIs; vortioxetine for the Serotonin Modulator and Stimulator antidepressants (SMS); bupropion, mirtazapine, reboxetine, and trazodone for the atypical antidepressants; amitriptyline, clomipramine, imipramine, and nortriptyline for the TCAs. Only English language studies were included.

### 2.3. Principles of Medicine Transfer from the Mother to the Infant through Human Milk

There are some methods to calculate the extent of drugs to which infants are exposed through breastmilk during lactation, such as the *breastmilk/plasma (M/P ratio)* and the *relative infant dose (RID)*. A significant excretion of the drug in milk, and therefore ingestion by the newborn, may increase the risk of actual intoxication of the breastfed infant. The *(M/P) ratio* is a measure that compares milk with maternal plasma drug concentrations. An M/P ratio <1.0 is indicative of the drug’s safety during breastfeeding.

When the concentration of a drug in breastmilk is known, it is possible to calculate the expected daily-ingested dose using the following formula: infant daily dose = concentration of the drug into the milk × total volume of ingested milk/day (presuming that an infant receives about 150 mL/kg per day). Knowing the infant daily dose, it was possible to calculate the RID by using the following formula: RID = infant daily dose (mg/kg/day) divided by the maternal dose of the drug (mg/kg/day) expressed as a percentage. A RID lower than 10% of the maternal weight-adjusted dosage is used as an arbitrary cut-off for medicine to be considered compatible with breastfeeding [6].

## 3. Results

Overall, 529 records were identified about the use of antidepressants during lactation. After the exclusion of papers published in languages other than English and experimental studies in animals, 420 studies were considered eligible. Searching only the full-text articles, 373 records were selected. A total of 133 papers were excluded concerning single case-reports and case-series with missing data about drug dosage. From the remaining records, 25 reports were not retrieved. The research strategy allowed the final identification of 215 references (Figure 1), and five recommendations were generated.

In particular, according to the four areas of investigation proposed by the consensus group, we identified 52 references about the assessment of the risks associated with untreated maternal depression during breastfeeding; after exclusion of repetitions and records regarding other psychiatric conditions, we found ten eligible papers. For the assessment of the risk of adverse events in breastfed infants associated with antidepressant and anxiolytic drug use during breastfeeding, including the transfer of the drugs from the mother to the infant through human milk, we selected 142 records. Twenty publications have been detected regarding the long-term developmental outcomes of infants’ cognitive development after maternal use of the drugs during lactation. About the evaluation of pharmacological treatment of opioid abuse in breastfeeding women with depressive disorders, we identified six studies. The literature results about the maternal, breastmilk, and infant drug concentrations and the effects on breastfed infants are summarized in the tables below.

### 3.1. Antidepressants

We found one meta-analysis and 69 case-series studies or case-reports concerning SSRIs, 14 case-series studies or case-reports concerning SNRIs, 19 case-series studies or case-reports on tricyclic antidepressants, and 14 case-series studies or case-reports concerning atypical antidepressants. Finally, the BDZs and HBRAs accounted for 25 case-series studies or case-reports.

#### 3.1.1. Maternal Plasma Levels, Breastmilk Levels, and Milk to Plasma Ratio

Regarding the antidepressants, we found 24 case-series and case-report studies that assessed the maternal plasma concentrations in relation to the breastmilk drug concentrations. Amitriptyline, fluoxetine, and paroxetine breastmilk concentrations were lower than in maternal plasma; much lower than in the maternal plasma were also the breastmilk concentrations of bupropion, reboxetine, and trazodone. All the other TCAs and SSRIs concentrations were higher in the breastmilk. The M/P ratio was higher than one for most of the antidepressants evaluated except for fluoxetine, paroxetine, bupropion, reboxetine, and trazodone (Table 1).

#### 3.1.2. Drug Concentrations into the Infant’s Plasma as a Measure of Drug Exposure via Milk

In Table 2 we reported the drug concentrations into the infant’s plasma as a measure of drug exposure via milk and the relative infant dose. We found undetectable to very low infant plasma concentrations for most antidepressants except for fluoxetine (detectable levels) and trazodone and mirtazapine (one study each).

#### 3.1.3. Effects on Breastfed Infants of Drug Exposure via Milk

Finally, we reported the effects on breastfed infants of drug exposure via milk and the number of cases published (Table 3). Twelve studies were found for tricyclic antidepressants for a total of 61 neonates exposed to TCAs during breastfeeding (twenty-four neonates for amitriptyline, seventeen for nortriptyline, eight for clomipramine, and twelve for imipramine). Only one case of sedation was reported.

Regarding the SSRI antidepressants, we identified 62 studies for a total of 644 neonates, of which 18 had symptoms that probably related to the medicine exposure through breastfeeding. Ten studies were found for citalopram for a total of 59 neonates exposed to citalopram during breastfeeding (six with presented symptoms); six for escitalopram for a total of 17 infants exposed (one with symptoms); eighteen studies for fluoxetine with 225 cases (five symptomatic); eight for fluvoxamine for a total of 13 infants exposed (one with symptoms); twelve studies were selected for paroxetine for a total of 116 cases exposed (three with symptoms); and finally, eighteen studies for sertraline with 214 infants exposed (two with symptoms). Eleven studies were selected for SNRIs: three studies for duloxetine (eight infants exposed, none with symptoms), while for venlafaxine there were four case-reports and four prospective cohort studies for a total of 46 infants exposed during breastfeeding (one with symptoms). Regarding the SMS antidepressant group, only one study is published to date concerning vortioxetine exposure in three infants without any adverse effect. Also, the atypical antidepressants have received little attention in the literature. Bupropion has been evaluated in four studies with five neonates exposed (two with symptoms) while mirtazapine, reboxetine, and trazodone have been checked in five (with sixty-two infants exposed), two (nine neonates exposed), and one (one neonate exposed) study, respectively; none of the infants had symptoms.

### 3.2. Benzodiazepines and Other Hypnotics

#### 3.2.1. Maternal Plasma Levels, Breastmilk Levels, and Milk to Plasma Ratio

Concerning benzodiazepines and other hypnotics, we reviewed 22 case-series and case-report studies assessing the maternal plasma concentrations compared to breastmilk drug levels. The breastmilk concentrations of alprazolam, clonazepam, clotiazepam, diazepam, and etizolam were lower than in maternal plasma; lorazepam, midazolam, and oxazepam were all significantly lower in breastmilk than in the maternal plasma. In one study, lormetazepam levels in breastmilk were higher than those in maternal plasma levels [108]. Most benzodiazepines had an M/P ratio below one; only lormetazepam [108] and nitrazepam [109] had an M/P ratio greater than one. Among HBRAs, zolpidem had the lower M/P ratio.

#### 3.2.2. Drug Concentrations into the Infant’s Plasma as a Measure of Drug Exposure via Milk

Furthermore, Table 4 also displays the concentrations of benzodiazepines and HBRAs in the infant’s plasma as a measure of drug exposure through milk and the relative infant dose. The infant’s plasma levels were evaluated in seventeen studies. Of these, very low or undetectable plasma concentrations were reported. The relative infant dose was low (<10%) in fourteen studies.

#### 3.2.3. Effects on Breastfed Infants of Drug Exposure via Milk

In Table 5, we reported the effects of benzodiazepines exposure via milk on breastfed infants and the number of cases published. Thirteen studies were identified for benzodiazepines and HBRAs for a total of 145 neonates exposed to benzodiazepines and HBRAs during breastfeeding (eleven neonates to alprazolam, two to brotizolam, twenty-three to clonazepam, one to clotiazepam, ten to diazepam, sixty-seven to lorazepam, five to lormetazepam, nineteen to midazolam, three to oxazepam, and four to zolpidem). Two studies [122,123] reported drowsiness and sedation in two breastfed infants related to alprazolam. In two studies evaluating diazepam, only one report found that one out of ten infants had symptoms such as sedation and poor suctioning. In studies that evaluated other benzodiazepines, 130 infants were found to have no adverse effects and two infants were symptomatic.

### 3.3. International Guidelines for Management of Depression during Postpartum

Many countries have developed guideline recommendations for the treatment of postpartum depression (Table 6). Most of these guidelines agree on encouraging breastfeeding and advise psychotherapy as an initial treatment for mild to moderate depression. The American College of Obstetricians and Gynecologists, the British Columbia Reproductive Mental Health Program and Perinatal Services, and the Dutch Society of Obstetrics and Gynecology advise continuing antidepressants with the drug to which the patient responded during pregnancy while the Nordic Federation of Societies of Obstetrics and Gynecology advise switching to a more favorable antidepressant.

## 4. Discussion

Major depressive disorder (MDD) is defined by the DSM–5-TR [137] as a common and potentially severe mood disorder: diagnostic criteria require that the patient is experiencing, during the same 2-week period, five or more symptoms, at least one of which should be either (1) depressed mood or (2) markedly diminished interest or pleasure. Peripartum depression is considered when a patient has a major depressive episode along with the peri-partum onset, and it is not mentioned as a separate disease. It is defined as a major depressive episode with the onset during pregnancy or within 4 weeks of delivery. It is also reported that symptom manifestation begins during pregnancy in about a third of patients with postpartum depression [138] while the prevalence of postpartum depression is thought to be approximately 10–20% [1,139].

### 4.1. Untreated Postpartum Depression

Untreated postpartum depression poses a serious threat to the emotional well-being of the mother and her confidence and capacity to care for her infant [140,141]. Further, maternal depression also has adverse effects on infant behavioral, emotional, and cognitive development [142,143,144,145]. For these reasons, treating the patient with postpartum depression should be imperative [146]. In fact, both the World Health Organization [147] and the Task Force on Breastfeeding of the Italian Minister of Health [148] state that a screening for postpartum depression and anxiety using a validated instrument is recommended and should be accompanied by diagnostic and management services for women who screen positive. The main concern of treating the patient with postpartum depression with antidepressant drugs is the potential adverse effects on the nursing infant. For the assessment of neonatal exposure, it is necessary to evaluate the drug excretion into the breastmilk and the possible effects in the breastfed infants.

### 4.2. Assessment of Neonatal Exposure

#### 4.2.1. Drug Excretion into Breastmilk

The excretion of antidepressant and anxiolytic drugs into milk occurs by passive diffusion; the degree of the drug excretion depends on several factors such as the route of administration and timing of the dose, the absorption rate, the half-life of the drug and the peak serum time, the dissociation constant, and the volume of distribution. It depends also on drug concentrations in the blood as well as the drug’s chemical properties (molecular weight, plasma protein binding capacity, degree of lipophilicity, and ionization).

Further, the drug concentrations in the milk may vary not only from the parameters listed below, but also from the type of milk examined, whether it is foremilk or hind milk [19]. Foremilk is the milk available when the baby starts feeding and is mostly water combined with other nutrients while hind milk is the milk the baby gets at the end of a feed and it is highly fatty. For example, Yoshida et al. reported a mother taking 100 mg daily of amitriptyline and she had a foremilk level of 30 mcg/L and a hind milk level of 113 mcg/L [10].

A commonly used parameter that serves as an index of the extent of drug excretion in the milk is the milk to plasma (M/P) ratio, which compares milk with maternal plasma drug concentrations. An M/P ratio less than 1.0 is preferred because it indicates that drug transfer into breastmilk is relatively low. Therefore, when attempting to choose the safest SSRI for the nursing woman, an M/P ratio < 1.0 could be reasonable, but it is not supported by all authors [149]. The M/P ratio does not reflect the absolute amount of drug ingested per day and can therefore be misleading in a risk assessment of neonatal drug exposure. In fact, no studies have examined whether there is a correlation between the M/P ratio values and the clinical outcome of breastfed infants.

Better than M/P, the concentration of a drug in milk and the relative infant dose (RID) are the determinant of infant exposure, which in turn could be more useful to assess the safety [150]. The RID is a quantitative estimate, meaning the amount of the medicines ingested by the breastfed infants. In the literature, a RID lower than 10% is often used as an arbitrary cut-off where breastfeeding is deemed safe [67]. This is based on the number of adverse drug reaction reports in a literature search encompassing data on several drugs being an order of magnitude less than the weight-adjusted dose to the mother [151]. The 10% limit has subsequently also been accepted by organizations such as the American Academy of Pediatrics [152]. Other guidelines, such as the Danish expert guideline, used a more conservative arbitrary value of 5% for psychotropic drugs [153]. As we have seen in our results, the RID of most antidepressants is below the cut-off of 10% and even when considering the more conservative cut-off of 5%, all TCAs; fluvoxamine, paroxetine, and sertraline among the SSRIs; and duloxetine, vortioxetine, bupropion, and trazodone should all be considered compatible during breastfeeding.

Similar to the antidepressants, our findings showed that the RID of alprazolam, brotizolam, clonazepam, clotiazepam, diazepam, etizolam, flunitrazepam, lorazepam, lormetazepam, nitrazepam, oxazepam, zaleplon, zolpidem, and zopiclone is less than 10%, with most values even below 5%.

Eventually, the drug concentrations into the infant’s blood are the direct determinant, and an excellent objective measure, of drug exposure via milk. This measure could be used to estimate drug accumulation in breast milk and the degree of infant exposure to the drug, as well as the infant’s ability to metabolize and clear the drug [154]. Although routine infant serum sampling for drug concentration analysis is generally not recommended [155], it can be useful when the infant has symptoms potentially indicative of excessive drug exposure; in this case, however, the test is usually available only at a research laboratory [156].

Lastly, the drug levels should be measured by high-performance liquid chromatography or gas chromatography-mass spectroscopy. Drug concentrations measured by immunoassay only, which is an inaccurate method of quantification, cannot provide an overarching measure of the ability of medications to enter the infant’s circulation [157]. Despite all these considerations, the drug concentration in infant plasma is a more direct measure of infant exposure. Among the SSRIs, paroxetine, fluvoxamine, duloxetine, escitalopram, and sertraline produce essentially undetectable plasma levels. Citalopram levels have been measured in some infants, although they resulted usually in relatively low levels. Fluoxetine can produce detectable and maybe the highest infant plasma concentrations [34].

#### 4.2.2. Effects in Breastfed Infants

The main question on the use of antidepressant drugs during breastfeeding is the possible risk to the nursed children. This eventuality for drugs with very low to undetectable concentrations into the human milk is unlikely. Lee et al., in their study, reported that there were no significant differences in the frequency of symptoms reported in the infants between the groups of 43 infants whose mothers were treated with antidepressants and 31 infants of mothers not taking any medication, with the presence of signs in 6.9% and 3.2%, respectively [55]. Other reviews based on published case-reports and case-series do not indicate a substantial risk for the exposed infants [90,158]. Although the single case-reports are difficult to interpret with regard to causality, whether the relationship to drug exposure is causal or the suspected adverse events arose by coincidence, these single case-reports may suggest a possible or probable risk to safety. In this review, we found that the 3.2% and the 5.7% of the neonates exposed via breastmilk to antidepressants and benzodiazepines, respectively, were symptomatic. Moreover, the symptoms were mild (except maybe for bupropion) and all manifestations resolved without any sequela.

### 4.3. Long-Term Effects

Breastfeeding and antidepressant medications are a challenging issue for clinicians. Indeed, it is important to weigh the beneficial and health protective effects of breastfeeding, as well as its clinical impact on psychiatric disorders, against the potential effects of the medication reaching the organism of the lactating neonate via breastmilk.

Women with postpartum depression symptoms are at risk of early breastfeeding cessation [159]. Postpartum depression and anxiety symptoms negatively influence breastfeeding outcomes, and several studies reported an association between this disorder and breastfeeding duration and intensity (suboptimal breastfeeding) [160,161,162,163]. Moreover, postpartum depression and suboptimal breastfeeding are associated with negative health outcomes for both mothers and infants. Indeed, postpartum depression is associated with reduced mother–infant attachment [164], child development and behavior problems [165], and increased risk of suicide [5]. For infants, lack of breastfeeding is associated with increased risk of infections [166], all-cause mortality [167], and chronic diseases such as type 2 diabetes [168]. For mothers, suboptimal breastfeeding is associated with an increased risk of breast and ovarian cancers, type 2 diabetes [169], hypertension [170], and cardiovascular disease [171].

Prospective, randomized, and double-blind controlled studies investigating long-term effects of antidepressant exposure during lactation are lacking due to severe ethical issues. Still, there is evidence that untreated mental disorders in mothers have a negative impact on children’s development and increase the risk of mental disorders later in life [172]. A recent study focusing on the changes in concentrations of antidepressants and mood stabilizers during pregnancy and lactation in serum and breastmilk at different time points found no significant differences in the development of children in the first 12 months. In this study, high concentration–dose ratios in breastmilk were found for venlafaxine as well as lamotrigine, and low concentration–dose ratios were found for quetiapine and clomipramine. Similarly, clomipramine and quetiapine showed low milk/serum–penetration ratios [173]. In another prospective study enrolling 280 mothers on psychotropic monotherapy during lactation vs. a group of 152 non-exposed infants, no differences in the children’s neurodevelopment were found between the case group and control group [174].

It has been a long and winding road from the 1980s to 2023, from a sentence like “Drugs may have an adverse effect when ingested by a breastfed infant” to papers reporting modern strategies to remove the need for maternal medications as a barrier to breastfeeding [175,176,177]. As far as we know, evidence is accumulating on the safety of antidepressants during lactation; nevertheless, an individual risk–benefit assessment should always be performed, as inter-individual differences may have a substantial effect on the breastfeeding infant’s response to treatment.

### 4.4. Breastfeeding during Pharmacological Treatment of Opioid Use Disorder

Many authors reported that women who have postpartum depression are more at risk of experiencing substance abuse compared to women in the postpartum period without depressive symptoms. On the other hand, women who have a history of substance abuse are more likely to show postpartum depressive symptoms [178].

Breastfeeding in opioid-dependent women has many positive physical and behavioral health effects for mother and infant, including reducing the severity of the neonatal withdrawal syndrome. Therefore, mothers should be encouraged to breastfeed. However, some possible exceptions could be represented by the return to substances use and some drug-related infective diseases such as HIV infection or tuberculosis [179]. Opioid Agonist Treatment (OAT) with methadone, buprenorphine, or the association buprenorphine-naloxone represent the treatment of choice for women with Opioid Use Disorder during pregnancy and should be continued after delivery [180]. Whether it was necessary to increase the dosages of methadone and buprenorphine due to the increase in total body fluids, the daily dosages of OAT medications could be reviewed after delivery in order to reduce the possible side effects, e.g., sedation, nausea, vomiting [181]. Methadone is primarily metabolized in the liver and it is mainly excreted in urine, feces, and in a small amount in sweat and saliva. In breastmilk, the relative infant dose compared to mother’s dose per kilogram ranges from 0.5% to 9%, and this may contribute to the alleviation of neonatal withdrawal symptoms during breastfeeding [182]. Buprenorphine is primarily metabolized by hepatic pathways. The metabolites are excreted in the biliary system with enterohepatic cycling. Most of the drug is excreted in the feces, urine, and in small amounts in the breastmilk. Given that the infant swallows the milk, absorption of buprenorphine from breastmilk would be expected to be minimal. In fact, the relative infant dose compared to mother’s dose per kilogram ranges from 0.04% to 0.63% [182]. In conclusion, the use of methadone and buprenorphine in opioid-dependent breastfeeding women is associated with better maternal and neonatal outcomes compared with uncontrolled opioid misuse.

### 4.5. Current Guidelines for the Treatment of Depression during Postpartum

There are few national guideline recommendations for the treatment of postpartum depression published to date. Many countries have not established their clinical practice guidelines for peripartum depression, and for those available, the recommendations are not always uniform [183]. These guidelines may facilitate the decision of the physicians about the antidepressant treatment during lactation, by weighting the possible risk of exposure via breastfeeding against the potential adverse effects of the untreated maternal depression to both the mother and child. They should adhere to the quality criteria of the Appraisal of Guidelines for Research and Evaluation (AGREE) [184] instrument and use the methods proposed by the Grades of Recommendation, Assessment, Development, and Evaluation (GRADE) working group [185]. It is an excellent facilitation of clinical practice, although it is a long, expensive, and laborious process. Moreover, as pointed out by Santos et al., guidelines do not disclose recommendations on emerging clinical questions and on new available evidence [186]. On the other hand, a consensus statement and its recommendations are developed based on a collective opinion or consensus of the convened expert panel, and for this reason some authors excluded the consensus statements from their reviews [183,187]. But when the available evidence conflicts or is insufficient due to lack of high-quality evidence, the consensus is still a valid alternative and this is the case in the field of antidepressant treatment during breastfeeding, where the available data are very few, conflicting, and the clinical trials are lacking for ethical reasons [188].

### 4.6. Psychotherapy

Although psychotherapy is effective for the treatment of postpartum depression, whether delivered in group or individual format [189], and also is recommended as the first-line approach for perinatal women with a new mild to moderate depression episode [190], it is not widely available, particularly to people in rural areas, because of excessive costs or unprepared socio-cultural context for psychotherapy [191,192]. As a consequence, women with mild to moderate postpartum depression may receive neither antidepressants nor psychotherapy. For moderate to severe postpartum depression or even for a mild to moderate depressive episode when psychotherapy is not available, antidepressants are a crucial component of treatment, both to relieve acute symptoms and to prevent the recurrence of depression.

### 4.7. Nighttime Breastfeeding

Nighttime breastfeeding is important for several reasons: prolactin release (the hormone that is responsible for lactation) follows a circadian rhythm, with highest levels of secretion at night and in the early morning [193]; tryptophan (an essential amino acid and a precursor of serotonin, melatonin, and nicotinamide) presents high levels in breastmilk during the night [194]. Eventually, frequent milk removal of the breast, even during the night, optimizes breastmilk production and may prevent ductal engorgement [195].

On the other hand, disturbed sleep during the nighttime increases the risk of relapse in postpartum depression [196,197]. Therefore, it is important to ensure that mothers with a mental disorder have a good circadian rhythm and get a good night’s sleep. Ross et al. [198] suggest that good quality nighttime sleep is the key in the recovery process from postpartum depression. The healthcare professional, therefore, should enlist the help of family members so the new mother can get 7 to 8 h of uninterrupted sleep. The breastfeeding mother should be encouraged to express milk throughout the day for at least some of the nighttime feedings to allow other caregivers to feed the infant [198]. Eventually, fatigue and disturbed sleep from breastfeeding at night or early morning may negatively affect breastmilk production, as well as the mother–baby attachment and interactions, thereby delaying the development of the infants [199,200,201].

## 5. Limitations

The main limitation of making a statement on the use of drugs in maternal postpartum depression is represented by the small number of cases reported in the literature to date. There is also the possibility of missed studies in the review strategy. Furthermore, despite the extensive literature research, some recommendations in this review are still based on expert opinion and clinical experience.

Eventually, the guidelines can be viewed as an expert consultation that should be weighed in conjunction with other updated information and in the context of each individual patient–healthcare professional relationship.

## 6. Ethical Considerations

An excellent physician–patient relationship is based on mutual trust and accurate information. During a medical consultation, a woman in need of advice on the management of depressive or anxiety disorders should be informed clearly and possibly also with informative materials. The informed consent is a prerequisite to ensure an adequate shared decision. Regular follow-up visits should be offered, preferably with a multidisciplinary approach. The evidence clearly suggests that treating underlying conditions during breastfeeding is the most recommended choice.

Drug treatment is rarely related to an enhanced risk of neonatal complications, and even in this case, these complications usually resolve quickly and without sequelae. In our review, no ethics approval was sought and no informed consent was collected, as this review evaluated existing clinical data and published case-reports. The synthesis was not registered in PROSPERO.

## 7. Future Research

More data are needed to address risks for newborns related to breastfeeding while the mother is on treatment with antidepressants and/or anxiolytic drugs. A limited number of studies was found after an extensive review of the current literature. Also, the reported studies are mostly case-reports or case-series, and a lack of randomized clinical trials was found. Most of the studies found also do not control for underlying conditions in the mother such as depression or anxiety disorders, so indication bias needs to be considered carefully in future research. Future research in the field should also more clearly address long-term safety for children exposed to antidepressants or anxiolytics during breastfeeding, especially concerning neurodevelopmental outcomes.

## 8. Conclusions

Although several international guidelines exist on the treatment of depressive and anxiety disorders during breastfeeding, our consensus is the first one in Italy. Our study supports a more optimistic attitude of health practitioners in respect of the patient affected by depression. First and foremost, the healthcare professional should be aware of the information already collected and personally processed by the mother and of her feeding plan for her baby. Secondly, the health professional should discuss with the patient the risks of untreated illness, the risks and benefits of treatment for the nursed infant, and the benefits of breastfeeding. Incorrect advice from physicians has a significant influence on the woman’s decision not to breastfeed even in the case of drugs compatible with nursing. In conclusion, the recommendations engendered are described in Table 7.

## Figures and Tables

**Figure 1 ijerph-21-00551-f001:**
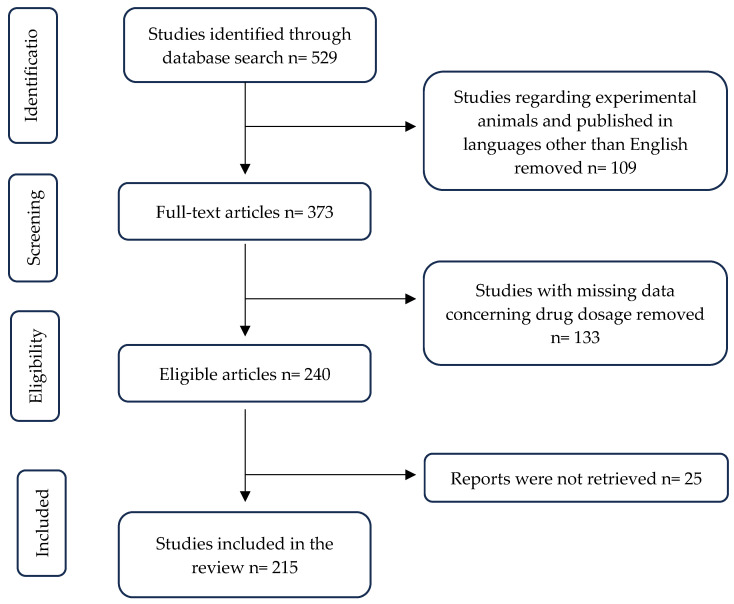
Flow chart of the identification and selection of evaluated studies.

**Table 1 ijerph-21-00551-t001:** Antidepressants: maternal plasma levels, breastmilk levels, and ratio of milk to plasma level (M/P).

		Maternal Plasma Levels (ng/mL)	Breastmilk Levels (ng/mL)	Ratio of Milk to Plasma Level (M/P)
Tricyclic antidepressants				
Amitriptyline	Weissman 2004 [7]	41–228	38–143	0.0–1.6
Nortriptyline	Matheson 1988 [8]	104–298	90–404	0.8–1.3
Clomipramine	Schimmell 1991 [9]	208.4–509.8	215.8–624.2	1.0–1.2
Imipramine	Yoshida 1997 [10]	29–340	110–610	1.8–3.8
SSRI				
Citalopram	Heikkinen 2002 [11]	40.3–64.6	90.3–144.5	2.2–2.2
	Berle 2004 [12]	18.5–164.9	51.0–235.4	1.4–2.7
Escitalopram	Castberg 2006 [13]	10.0–37.5	24.9–76.1	2.0–2.4
	Rampono 2006 [14]	9.0–49.0	27.0–99.0	2.0–3.0
Fluoxetine	Kristensen 1999 [15]	108–809	54–705	0.5–0.8
	Hendrick 2001 [16]	64.0–1180	49.0–457	0.3–0.7
Fluvoxamine	Kristensen 2002 [17]	29.7–190.8	35.9–256.3	1.2–1.3
Paroxetine	Begg 1999 [18]	32.2–117.8	19.8–61.7	0.5–0.6
	Ohman 1999 [19]	16–164	8–92	0.5–0.5
Sertraline	Kristensen 1998 [20]	15.7–92.0	27.9–193.3	1.7–2.1
	Schoretsanitis 2019 [21]	3.3–17.8	3.6–35.7	1.0–2.0
SNRI				
Venlafaxine	Ilett 2002 [22]	264.9–600.7	856.0–1464.0	2.4–3.2
	Newport 2009 [23]	292.0–1210	739.0–2231.3	1.8–2.5
Duloxetine	Briggs 2009 [24]	24.0–53.0	31.0–64.0	1.2–1.2
	Collin-Lévesque 2018 [25]	39.7–60.6	14.3–29.3	0.3–0.4
SMS				
Vortioxetine	Marshall 2021 [26]	No data	13.89–52.32	No data
Atypical antidepressants				
Bupropion	Davis 2009 [27]	13.5–150.0	8.5–11.5	0.0–0.6
Mirtazapine	Kristensen 2007 [28]	24.0–70.0	42.0–65.0	0.7–1.5
Reboxetine	Hackett 2006 [29]	115–321	6.7–16.3	0.0–0.0
Trazodone	Saito 2021 [30]	69.3–267.6	18.2–50.2	0.2–0.2

**Table 2 ijerph-21-00551-t002:** Antidepressants: infant plasma levels and relative infant dose.

		Infant Plasma Levels (ng/mL)	Relative Infant Dose (RID) *
Tricyclic antidepressants			
Amitriptyline	Bader 1980 [31]	Undetectable	1.9–2.8%
Nortriptyline	Weissman 2004 [7]	Undetectable	1.7–3.1%
Clomipramine	Yoshida 1997 [10]	Undetectable	2.8%
SSRI			
Citalopram	Heikkinen 2002 [11]	Undetectable to very low	3.56–5.37%
Escitalopram	Rampono 2006 [14]	Undetectable	-
	Delaney 2018 [32]	Undetectable	5.2–7.9%
Fluoxetine	Lester 1993 [33]	Detectable	-
	Epperson 2003 [34]	Detectable	-
	Weissman 2004 [7]	Detectable	1.6–14.6%
	Keymer 2023 [35]	Detectable	-
Fluvoxamine	Kristensen 2002 [17]	Undetectable to very low	0.3–1.4%
	Hendrick 2001 [16]	Undetectable to very low	-
Paroxetine	Berle 2004 [12]	Undetectable to very low	1.2–2.8%
Sertraline	Stowe 2003 [36]	Undetectable to very low	0.4–2.2%
SNRI			
Venlafaxine	Ilett 2002 [22]	Undetectable to very low	6.8–8.1%
	Newport 2009 [23]	Undetectable to very low	-
Duloxetine	Briggs 2009 [24]	Undetectable	0.1–1.1%
	Boyce 2011 [37]	Undetectable	-
SMS			
Vortioxetine	Marshall 2021 [26]	No data	1.1–1.7%
Atypical antidepressants			
Bupropion	Baab 2002 [38]	Undetectable to very low	0.2–2%
	Neuman 2014 [39]	Undetectable to very low	-
Mirtazapine	Klier 2007 [40]	Undetectable	-
	Kristensen 2007 [28]	Undetectable	1.6–6.3%
	Tonn 2009 [41]	Detectable	-
Reboxetine	Hackett 2006 [29]	Undetectable to very low	No data
Trazodone	Saito 2021 [30]	Detectable	2.8%

* From Hale’s Medications and Mothers’ Milk 2023: A Manual of Lactational Pharmacology. Springer Publishing 20th Edition [42].

**Table 3 ijerph-21-00551-t003:** (**A**) Tricyclic Antidepressants and effects on breastfed infants. (**B**) SSRI antidepressants and effects on breastfed infants. (**C**) SNRI and SMS antidepressants and effects on breastfed infants. (**D**) Atypical antidepressants and effects on breastfed infants.

Drug	Study	Effects on Breastfed Infants	Number of Infants
**A**			
Amitriptyline	Misri 1991 [43]	No adverse effects	20
	Breyer-Pfaff 1995 [44]	No adverse effects	1
	Yoshida 1997 [10]	No adverse effects	2
	Uguz 2017 [45]	Sedation	1
Nortriptyline	Matheson 1988 [8]	No adverse effects	1
	Wisner 1991 [46]	No adverse effects	7
	Wisner 1997 [47]	No adverse effects	6
	Birnbaum 1999 [48]	No adverse effects	3
Clomipramine	Wisner 1995 [49]	No adverse effects	4
	Yoshida 1997 [10]	No adverse effects	2
	Birnbaum 1999 [48]	No adverse effects	2
Imipramine	Yoshida 1997 [10]	No adverse effects	4
	Birnbaum 1999 [48]	No adverse effects	2
	Uguz 2016 [50]	No adverse effects	6
**B**			
Citalopram	Jensen 1997 [51]	No adverse effects	1
	Spigset 1997 [52]	No adverse effects	3
	Rampono 2000 [53]	No adverse effects	7
	Schmidt 2000 [54]	Irritability, restlessness	1
	Heikkinen 2002 [11]	No adverse effects	11
	Lee 2004 [55]	No adverse effects	28
	Lee 2004 [55]	Irritability	3
	Franssen 2006 [56]	Irritability	1
	Werremeyer 2009 [57]	No adverse effects	1
	Pogliani 2019 [58]	No adverse effects	2
	Akbarzadeh 2023 [59]	Bruxism	1
Escitalopram	Castberg 2006 [13]	No adverse effects	1
	Gentile 2006 [60]	No adverse effects	1
	Hackett 2006 [29]	No adverse effects	1
	Ilett 2005 [61]	No adverse effects	5
	Rampono 2006 [14]	No adverse effects	8
	Potts 2007 [62]	Necrotizing enterocolitis	1
Fluoxetine	Moretti 1989 [63]	No adverse effects	45
	Taddio 1996 [64]	No adverse effects	11
	Kim 1997 [65]	No adverse effects	6
	Yoshida 1998 [66]	No adverse effects	4
	Birnbaum 1999 [48]	No adverse effects	13
	Chambers 1999 [67]	No adverse effects	26
	Kristensen 1999 [15]	No adverse effects	8
	Kristensen 1999 [15]	Hyperactivity, colic	2
	Hale 2001 [68]	Serotonin syndrome	1
	Hendrick 2001 [16]	No adverse effects	19
	Nulman 2002 [69]	No adverse effects	21
	Suri 2002 [70]	No adverse effects	10
	Epperson 2003 [34]	No adverse effects	11
	Heikkinen 2003 [71]	No adverse effects	11
	Oberlander 2005 [72]	No adverse effects	6
	Kim 2006 [73]	No adverse effects	27
	Morris 2015 [74]	Serotonin syndrome	1
	Pogliani 2019 [58]	No adverse effects	2
	Keymer 2023 [35]	Serotonin syndrome	1
Fluvoxamine	Yoshida 1997 [75]	No adverse effects	1
	Arnold 2000 [76]	No adverse effects	1
	Hägg 2000 [77]	No adverse effects	1
	Hendrick 2001 [16]	No adverse effects	4
	Kristensen 2002 [17]	No adverse effects	2
	Gentile 2006 [60]	No adverse effects	1
	Piontek 2001 [78]	No adverse effects	2
	Uguz 2015 [79]	Diarrhea, vomiting, agitation	1
Paroxetine	Spigset 1996 [76]	No adverse effects	1
	Begg 1999 [18]	No adverse effects	10
	Birnbaum 1999 [48]	No adverse effects	1
	Ohman 1999 [19]	No adverse effects	6
	Hendrick 2000 [80]	No adverse effects	1
	Misri 2000 [81]	No adverse effects	24
	Stowe 2000 [80]	No adverse effects	16
	Hendrick 2001 [16]	No adverse effects	16
	Merlob 2004 [82]	No adverse effects	26
	Merlob 2004 [82]	Irritability	1
	Uguz 2018 [83]	Constipation	1
	Pogliani 2019 [58]	No adverse effects	5
	Uguz 2019 [84]	No adverse effects	7
	Uguz 2019 [84]	Restlessness	1
Sertraline	Altshuler 1995 [85]	No adverse effects	1
	Mammen 1997 [86]	No adverse effects	3
	Epperson 1997 [87]	No adverse effects	4
	Stowe 1997 [88]	No adverse effects	9
	Kristensen 1998 [20]	No adverse effects	8
	Wisner 1998 [89]	No adverse effects	8
	Birnbaum 1999 [48]	No adverse effects	3
	Dodd 2000 [90]	No adverse effects	10
	Holland 2000 [91]	No adverse effects	6
	Epperson 2001 [92]	No adverse effects	19
	Hendrick 2001 [16]	No adverse effects	33
	Stowe 2003 [36]	No adverse effects	26
	Pearlstein 2006 [93]	No adverse effects	12
	Wisner 2006 [94]	No adverse effects	55
	Hantsoo 2014 [95]	No adverse effects	6
	Uvais 2017 [96]	Diarrhea	1
	Uguz 2018 [83]	Restlessness	1
	Pogliani 2019 [58]	No adverse effects	9
**C**			
Venlafaxine	Ilett 1998 [97]	No adverse effects	3
	Hendrick 2001 [98]	No adverse effects	2
	Ilett 2002 [22]	No adverse effects	7
	Newport 2009 [23]	No adverse effects	13
	Pogliani 2019 [58]	No adverse effects	1
	Rampono 2011 [99]	No adverse effects	10
	Baldelli 2022 [100]	No adverse effects	1
	Eleftheriou 2022 [101]	Sedation and hypotonia	1
Duloxetine	Briggs 2009 [24]	No adverse effects	1
	Lobo 2008 [102]	No adverse effects	6
	Boyce 2011 [37]	No adverse effects	1
Vortioxetine	Marshall 2021 [26]	No adverse effects	3
**D**			
Bupropion	Briggs 1993 [103]	No adverse effects	1
	Baab 2002 [38]	No adverse effects	2
	Chaudron 2004 [104]	Seizures	1
	Neuman 2014 [39]	Abnormal movements	1
Mirtazapine	Aichhorn 2004 [105]	No adverse effects	1
	Klier 2007 [40]	No adverse effects	1
	Kristensen 2007 [28]	No adverse effects	8
	Smit 2015 [106]	No adverse effects	44
	Uguz 2019 [84]	No adverse effects	8
Reboxetine	Hackett 2006 [29]	No adverse effects	4
	Berlin 2019 [107]	No adverse effects	5
Trazodone	Saito 2021 [30]	No adverse effects	1

**Table 4 ijerph-21-00551-t004:** Benzodiazepines and HBRAs: maternal plasma levels, breastmilk levels, ratio of milk to plasma level (M/P), infant plasma levels, and relative infant dose (RID).

Drug	Study	Maternal Plasma Levels (ng/mL)	Breastmilk Levels (ng/mL)	Ratio of Milk to Plasma Level (M/P)	Infant Plasma Levels (ng/mL)	Relative Infant Dose (RID) %
Alprazolam						
	Oo 1995 [110]	6.19–11.57	2.11–5.29	0.36	-	3
	Furugen 2019 [111]	13.3	5.42	0.41	-	3.11–4.61
	Nishimura 2021 [112]	13.3	5.42	0.41	-	3.8–9.3
	Saito 2022 [113]	1.0–2.3	0.5–0.9	0.21–0.9	-	1.4
Brotizolam						
	Nishimura 2021 [112]	0.589	0.272	0.59	-	2.3
	Saito 2021 [114]	0.51	0.12	0.24	-	-
	Saito 2022 [113]	0.3–5.0	Not detected	-	-	-
Clonazepam						
	Fisher 1985 [115]	32	11–13	0.33	4.4	-
	Nishimura 2021 [112]	15	6.07	0.17	-	4.6
Clotiazepam						
	Nishimura 2021 [112]	109	16.3	0.15	-	2.5
Diazepam						
	Dusci 1990 [116]	89–2500	6–307	0.17–0.23	<5	-
	Hale 2012 [117]	-	-	-	-	7.1
Etizolam						
	Nishimura 2021 [112]	4.63	0.77	0.17	-	0.6
Flunitrazepam						
	Nishimura 2021 [112]	1.73	1.19	0.69	-	1.6–2.5
Lorazepam						
	Summerfield 1985 [118]	38	9	0.24	-	-
	Nishimura 2021 [112]	6.78–7.94	1.27–1.5	0.16–0.21	-	2.1–4.4
Lormetazepam						
	Humpel 1982 [119]	3.5	<0.2	0.06	<0.09	0.35
	Lemmer 2007 [108]	3.1	5.3	1.7	-	-
Midazolam						-
	Matheson 1990 [109]	30.9–33.3	<10	0.3	-	-
Nitrazepam						
	Matheson 1990 [109]	0.17–0.21	0.22–0.34	1.3–1.6	<2.8	2.6
Oxazepam						
	Dusci 1990 [116]	-	Not Detected to 30	0.09–0.1	7.5–9.6	-
	Lebedevs 1992 [120]	<5–9	Not Detected		Not Detected	
Zolpidem						
	Pons 1989 [121]	90–364	0.76–3.88	0.002–0.04	-	0.004–0.019

**Table 5 ijerph-21-00551-t005:** Benzodiazepines and other hypnotics: effects on breastfed infants.

Drug	Study	Effects on Breastfed Infants	Number of Infants
Alprazolam			
	Ito 1993 [122]	Drowsiness (1)	5
	Kelly 2012 [123]	Sedation (1)	6
Brotizolam			
	Nishimura 2021 [112]	No adverse effects	1
	Saito 2022 [113]	No adverse effects	1
Clonazepam			
	Fisher 1985 [115]	Apnea, cyanosis at birth	1
	Kelly 2012 [123]	Sedation (1)	22
Clotiazepam			
	Nishimura 2021 [112]	No adverse effects	1
Diazepam			
	Wesson 1985 [124]	Sedation, poor suctioning	1
	Kelly 2012 [123]	No adverse effects	9
Lorazepam			
	Kelly 2012 [123]	No adverse effects	64
	Nishimura 2021 [112]	No adverse effects	3
Lormetazepam			
	Humpel 1982 [119]	No adverse effects	5
Midazolam			
	Kelly 2012 [123]	No adverse effects	19
Oxazepam			
	Kelly 2012 [123]	No adverse effects	2
	Dusci 1990 [116]	No adverse effects	1
Zolpidem			
	Saito 2022 [125]	No adverse effects	1
	Saito 2022 [126]	No adverse effects	3

In parenthesis, the number of breastfed infants with symptoms.

**Table 6 ijerph-21-00551-t006:** International guidelines for management of depression in breastfeeding mothers.

Scientific Society	Country	Main Recommendations
National Institute for Health and Care Excellence, 2018 [127]	United Kingdom	Advise psychotherapy and pharmacologic treatment for new-onset depression
Royal Australian and New Zealand College of Psychiatrists, 2015 [128]	Australia and New Zealand	Advise psychotherapy and pharmacologic treatment for new-onset depression
Canadian Network for Mood and Anxiety Treatments, 2016 [129]	Canada	Advise psychotherapy and pharmacologic treatment for new-onset depression
Scottish Intercollegiate Guidelines Network, 2012 [130]	United Kingdom	Advise psychotherapy and pharmacologic treatment for new-onset depression
American College of Obstetricians and Gynecologists, 2023 [131]	USA	Advise psychotherapy and pharmacologic treatment for new-onset depression; continue previous antidepressant therapy
Nordic Federation of Societies of Obstetrics and Gynecology, 2015 [132]	Norway	Advise switching antidepressants when unfavorable medication is used during breastfeeding
Dutch Society of Obstetrics and Gynecology, 2012 [133]	Netherlands	Advise continuation of antidepressant medication to prevent relapse of depressive symptoms; continue previous antidepressant therapy
British Columbia Reproductive Mental Health Program & Perinatal Services, 2014 [134]	Canada	Advise continuation of antidepressant medication to prevent relapse of depressive symptoms; continue previous antidepressant therapy
Polish Psychiatric Association, 2019 [135]	Poland	Advise psychotherapy and pharmacologic treatment for new-onset depression
British Association for Psychopharmacology, 2019 [136]	United Kingdom	Advise pharmacologic treatment for new-onset depression; continue previous antidepressant therapy

**Table 7 ijerph-21-00551-t007:** Consensus recommendations.

Areas of Investigation	Main Recommendations
Risk of untreated depression	Untreated postpartum depression poses a serious threat to the emotional well-being of the mother and her confidence and capacity to care for her infant. Recommendation 1. Depressive and anxiety disorders are not a contraindication for breastfeeding, nor is a pharmacological treatment. In the case of discontinuation of pharmacological treatment, the Panel recommends that treatment should not be stopped abruptly, in order to avoid relapse.
Risk of adverse events in breastfed infants associated with antidepressant and anxiolytic drug use during breastfeeding	As for all drug use during breastfeeding, possible neonatal risk cannot be excluded; the risks are limited/very limited but not zero. Recommendation 2. Patients with depression who require standard treatments to control their disease must be followed-up closely by the psychiatrist. The same should be done for their infants by the pediatrician. The Panel recommends that cooperation between specialists (e.g., psychiatrists, pediatrician, toxicologist) should be encouraged. Coordination with the general practitioner is also recommended. The Panel recommends offering multidisciplinary care whenever possible. Recommendation 3. The Panel recommends continuing the medication to which the patient has responded well during pregnancy. When there is the need to start an antidepressant during breastfeeding, drugs with a more favorable safety profile and more epidemiological data, such as the SSRI, should be preferred and prescribed at the lowest effective dose. According to current studies, sertraline, escitalopram, and paroxetine appear to be the safest among SSRIs. They are the most highly rated antidepressants, with a RID < 10% and very rare and mild adverse effects reported to the breastfeed infants. Recommendation 4. For the treatment of anxiety symptoms and short-term treatment of sleep disturbances, benzodiazepines can be administered during breastfeeding. In order to minimize the pharmacological effects of the drug on the infant, it is preferable to avoid drugs with multiple active metabolites and a long half-life (e.g., diazepam), and opt for drugs with a shorter elimination half-life, such as lorazepam, oxazepam, and brotizolam. The Panel concluded that benzodiazepines could be used during breastfeeding and recommends using short half-life drugs.
Long-term developmental outcomes of infants’ cognitive development after maternal use of the drugs during lactation	The long-term effects on a child’s neurological development cannot be excluded, although such risk is hypothetical, still unproven, and unlikely given the exposure to very small doses of psychotropic drugs through breastmilk.
Evaluation of pharmacological treatment of opioid abuse in breastfeeding women with depressive disorders	Women with postpartum depression are more at risk of experiencing substance abuse and women who have a history of substance abuse are more likely to show postpartum depressive symptoms. Recommendation 5. In opioid-dependent breastfeeding women, the use of methadone and buprenorphine is associated with better maternal and neonatal outcomes compared with uncontrolled opioid misuse. There are no recommended pharmacological interventions for the abuse of psychoactive stimulants, cannabinoids, or the new psychoactive substances during breastfeeding. The Panel recommends that opioid abuse treatment should not be stopped.

## Data Availability

No new data were created or analyzed in this study. Data sharing is not applicable to this article.

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
