# Peer review of "Consensus Panel Recommendations for the Pharmacological Management of Breastfeeding Women with Postpartum Depression"

_ijerph, 2024, doi:10.3390/ijerph21050551_

Round 1
Reviewer 1 Report
Comments and Suggestions for Authors
Dear Editor,
Please find below my comments on the article titled "Consensus Panel Recommendations for the Pharmacological Management of Breastfeeding Women with Postpartum depression".
The issue of medication use, particularly during pregnancy and lactation, presents a significant challenge for clinicians. Hence, I recognize the importance of the study presented by the authors.
I have attached sticky notes to the document highlighting the significant key points regarding the study. While the study is interesting overall, some sections require restructuring within the text. I believe there is a need for significant revisions in the presentation of the methods and in the introduction section. I think this manuscript requires major revision before being published in IJERPH.
Moreover, I believe it is imperative to address the plagiarism concern identified in the article before delving into the study design, obtained results, and the discussion points raised by the authors. As per the received report, approximately 40% of the article exhibits plagiarism.
Sincerely,

Author Response
Reviewer 1
- I have attached sticky notes to the document highlighting the significant key points regarding the study. PDF with suggestions attached.
We thank the reviewer for the suggestions. We attached the pdf file with all the corrections asked by the reviewer.
- While the study is interesting overall, some sections require restructuring within the text. I believe there is a need for significant revisions in the presentation of the methods and in the introduction I think this manuscript requires major revision before being published in IJERPH.
Please, find attached the new paper with the revisions in the presentation of the methods and introduction as well as the shorten discussion
- Moreover, I believe it is imperative to address the plagiarism concern identified in the article before delving into the study design, obtained results, and the discussion points raised by the authors. As per the received report, approximately 40% of the article exhibits plagiarism.
We have sent to the assistant editor the clarification a couple of weeks ago. Indeed, there are many similarities in this paper with our previous work, regarding the abstract, introduction, materials and methods section. This review is the second part of the “consensus panel recommendations for the pharmacological management of women with depression”. The focus of the first one was the pregnancy and the second one is breastfeeding. It could be possible to describe the methods “as reported previously” and cite my first paper, but from the other hand, this cannot help either the reviewers to suggest any improvement or the readers. Eventually, I think that it is not ethic the self-citation if not strictly necessary.
From the other hand, we absolutely agree that 40% of plagiarism is too high, so we have modified all sections.

Reviewer 2 Report
Comments and Suggestions for Authors
The work behind this manuscript should be commended. The ability to engage the different associations and the scientific community was impressive.
However, the report is too extensive and presented in a confusing way.
Abstract- The abstract does not reflect or explicitly say the 5 recommendations are provided (concise guide); this is an important contribution that should be in the abstract.
Materials and methods- I recommend dividing the methodology in two distinct and labeled sections: 2.1- Establishing Consensus and 2.2 Literature Search Strategy. Also specify in the Literature search strategy that the result of the first NGT informed the literature search. It will be good to have a timeline in which the reader sees the three steps clearly with the outcomes of each stage.
Results- Specify that this are the results of the literature review.
The literature review will be better understood and followed by the readers if it is organized following the four areas of investigation agreed by the consensus group.
Evaluate the need to include the tables as part of the narrative. For example, table 3 is summarized effectively starting in line 230. The table interrupts the reading and the flow and does not add information. It can be an appendix., as well as others.
Discussion- The discussion should also be organized by the four areas stated above and lead to the recommendations stated in the conclusion. There are missed opportunities in the discussion to better link to the concluding recommendations.
The conclusion- Very good and clearly stated.
Author Response
Reviewer 2
- The report is too extensive and presented in a confusing
We thank the reviewer for the suggestions. We have made many revisions in the presentation of the methods and introduction as well as the shorten discussion
- The abstract does not reflect or explicitly say the 5 recommendations are provided (concise guide); this is an important contribution that should be in the abstract.
We have changed the abstract and we have added the five recommendations as the reviewer asked to do.
- I recommend dividing the methodology in two distinct and labeled sections: 2.1- Establishing Consensus and 2.2 Literature Search Strategy.
We have divided the methodology in the sections the reviewer asked to do.
- Also specify in the Literature search strategy that the result of the first NGT informed the literature search. It will be good to have a timeline in which the reader sees the three steps clearly with the outcomes of each stage.
We added the three steps clearly with the outcomes of each stage as well the timeline with the outcomes of each stage.
- Results- Specify that this are the results of the literature review.
We specify that the results were obtained through an extensive literature review, in the first line of the results section
- The literature review will be better understood and followed by the readers if it is organized following the four areas of investigation agreed by the consensus group.
We modified the literature review and we have organized it following the four areas of investigation
- Evaluate the need to include the tables as part of the narrative. For example, table 3 is summarized effectively starting in line 230. The table interrupts the reading and the flow and does not add information. It can be an appendix, as well as others.
We agree that the tables maybe interrupt the reading, but I’m afraid that this is the format of the supplement of IJERPH. We will ask the assistant editor if it possible to add an appendix or to put all tables at the end of the paper
P.S. We have asked the assistant editor and it is possible to pit the tables in the appendix
- The discussion should also be organized by the four areas stated above and lead to the recommendations stated in the conclusion. There are missed opportunities in the discussion to better link to the concluding recommendations.
We have organized the discussion in the four areas stated above

Reviewer 3 Report
Comments and Suggestions for Authors
I am grateful for the possibility to carry out the following review and I congratulate the authors for their work, it is a very interesting work but I consider it necessary to make some modifications for its improvement:
Introduction: I think it would be necessary to formulate a research question and write a new objective, providing is not an appropriate verb when expressing an objective, it is more a goal of the authors, not the objective of the research.
Methodology: as this is a review study, there is a lack of a clear methodology when carrying out the study. I recommend conducting it following the PRISMA recommendations for systematic reviews, which will give more credibility to the results by conducting it in a systematic and reproducible way.
Results: I recommend adjusting the results to a specific methodology as mentioned above.
Author Response
Reviewer 3
- Introduction: I think it would be necessary to formulate a research question and write a new objective, providing is not an appropriate verb when expressing an objective, it is more a goal of the authors, not the objective of the research.
We thank the reviewer for the suggestions. We re-formulate and wrote the new objective using words more appropriate
- Methodology: as this is a review study, there is a lack of a clear methodology when carrying out the study. I recommend conducting it following the PRISMA recommendations for systematic reviews, which will give more credibility to the results by conducting it in a systematic and reproducible way.
In the new version, we followed the PRISMA recommendations (although our work is not a systematic review) and we added the flowchart of the literature search. Moreover, we have organized it following the four areas of investigation as suggested by the reviewer
- Results: I recommend adjusting the results to a specific methodology as mentioned above.
We have organized the result section following the four areas of investigation

Round 2
Reviewer 1 Report
Comments and Suggestions for Authors
Dear. Editor
I believe that the responses and revisions made by the authors in accordance with the suggestions I have made have considerably improved the overall quality of the study. I state that it is appropriate to be published in this form.
Sincerely regards
Reviewer 3 Report
Comments and Suggestions for Authors
The authors have made the necessary modifications so that the paper can be accepted.